

# Assessing population structure and morpho-molecular characterization of sunflower (*Helianthus annuus* L.) for elite germplasm identification

Sampath Lavudya[1], Kalaimagal Thiyagarajan[1], Sasikala Ramasamy[2], Harish Sankarasubramanian[2], Senthivelu Muniyandi[2], Anita Bellie[3], Sushil Kumar[4] and Susmitha Dhanapal[2]

[1] Department of Genetics and Plant Breeding, CPBG, Tamil Nadu Agricultural University, Coimbatore, Tamil Nadu, India
[2] Department of Oilseeds, CPBG, Tamil Nadu Agricultural University, Coimbatore, Tamil Nadu, India
[3] Department of Nematology, Tamil Nadu Agricultural University, Coimbatore, Tamil Nadu, India
[4] Department of Agricultural Biotechnology, Anand Agricultural University, Anand, Gujarat, India

Corresponding authors
Sampath Lavudya, sampathabhiram3@gmail.com
Kalaimagal Thiyagarajan, Kalaimagal.t@tnau.ac.in

## ABSTRACT

Sunflower (*Helianthus annuus* L.), known for its adaptability and high yield potential, is vital in global edible oil production. Estimating genetic diversity is a key pre-breeding activity in crop breeding. The current study comprised of 48 genotypes which were assessed for their biometrical traits at department of Oilseeds, Tamil Nadu Agricultural University, during the rainy season of 2022. The lines were subsequently characterised using 103 simple sequence repeat (SSR) markers for molecular diversity analysis. The results indicated that the net nucleotide distances indicated varying genetic divergence, with subpopulations II and V showing the highest (0.056) and I and IV the lowest (0.014). Subpopulation IV exhibited the highest heterozygosity (0.352), while subpopulation III had the lowest heterozygosity and a low Fst (0.173). Principal components analysis (PCA) and hierarchical cluster analysis were employed for assessing the morphological diversity, facilitating genotype grouping and parent selection for breeding programs. The first four components cumulatively accounted for 86.72% of the total variation. Cluster Analysis grouped 48 sunflower genotypes into three clusters based on genetic diversity. COSF 13B stands out for its high head diameter, oil content, seed yield, and oil yield based on mean performance of morphological data. Principal coordinate analysis (PCoA) mirrored the groupings from the Neighbor Joining method, with the first three components explaining 27.24% of the total variation. Molecular data analysis identified five distinct clusters among the germplasm. By integrating morphological and molecular marker data with genetic distance analysis, substantial diversity was revealed with the genotypes RHA 273 and GMU 325 consistently demonstrated high oil yield per plant. The genotypes GMU 477, GMU 450, COSF 13B, RHA 102, CMS 1103B, and RHA GPR 58 have been identified as suitable parents for enhancing oil content in sunflower breeding programs. These findings also aid in selecting SSR markers for genotype characterization and in choosing diverse parents for breeding programs.

## INTRODUCTION

Sunflower (*Helianthus annuus* L.) stands out as a significant oilseed crop worldwide, alongside groundnut, mustard and soybean. The genus is native to southwestern North America, and the cultivated sunflower was domesticated in central North America, it belongs to the Asteraceae family (*Ahmadpour et al., 2022*; *Reddy et al., 2024*). It is a highly cross-pollinated crop with $2n = 34$ chromosomes, sunflower demonstrates exceptional yield potential and adaptability to various environmental conditions (*Reddy et al., 2024*). Moreover, sunflower ranks as the world's fourth most crucial source of edible vegetable oil, contributing up to 12% of global edible oil production (*Dimitrijević et al., 2017*; *Rauf et al., 2017*).

Multivariate analysis encompasses statistical techniques used to analyze data arising from multiple variables. Among these techniques are principal component analysis (PCA) and cluster analysis. Cluster analysis is utilized to designate genetic diversity by examining resemblances or dissimilarities among genotypes. Meanwhile, PCA aims to eliminate redundancy in datasets and reveal patterns of distribution (*Dudhe et al., 2020*).

Evaluating genetic diversity through agro-morphological traits can be arduous and time-consuming, particularly without prior knowledge of variability and often proves unreliable due to significant environmental influences (*Ikram et al., 2020*; *Ibrar et al., 2022*). Yield, a complex quantitative trait, faces further complications due to its dependence on other variables, especially in crops like sunflower characterized by high cross-pollination and heterozygosity (*Arshad, Ilyas & Khan, 2007*; *Benchasri, Simla & Harakotr, 2020*; *Ahmad et al., 2021*; *Ibrar et al., 2022*).

A diverse germplasm collection is vital for crop improvement, facilitating efficient allele accumulation and reducing screening efforts (*Darvishzadeh et al., 2010*). In sunflower, genetic characterization is essential for diversity assessment, germplasm conservation, and marker-assisted selection. While morphological traits often fail to distinguish closely related cultivars, DNA markers provide reliable insights into polymorphism and genetic diversity, unaffected by environmental factors or epistasis (*Zeinalzadeh-Tabrizi et al., 2018*). Estimating genetic diversity using molecular markers is crucial in sunflower breeding, where accurate identification of parental lines is key (*Darvishzadeh et al., 2010*; *Zeinalzadeh-Tabrizi et al., 2018*; *Ahmed et al., 2022*).

Morphological characterization, biochemical and molecular marker techniques are used to assess genetic diversity, but the former two are less reliable due to environmental effects (*Rani et al., 2023*). DNA markers, unaffected by environmental factors, are essential for understanding genetic diversity (*Kumar et al., 2021*; *Rani et al., 2023*). Molecular markers like random amplified polymorphic DNA (RAPD), simple sequence repeat (SSR), Expressed Sequence Tag-Simple Sequence Repeat (EST-SSR), amplified fragment length polymorphism (AFLP), genotyping-by-sequencing (GBS) and single nucleotide polymorphism (SNP) have been employed to assess the genetic diversity in sunflower

germplasm. SSR markers due to their genome-wide distribution, high polymorphic information content, and reproducibility proved to be valuable for analyzing genetic diversity, and linkage mapping (*Guan et al., 2010*). Single nucleotide polymorphisms (SNPs) are stable and ideal for complex trait analysis but are less preferred for genetic diversity studies due to their limited information content, biallelic nature, and higher cost (*Alemu et al., 2020*). Population structure (PS) assessment often relies on SSR markers due to their established utility and higher information content compared to biallelic markers (*Dreisigacker et al., 2005*; *Filippi et al., 2015*). Several authors have assessed diversity and population structure in sunflower using SSR markers (*Filippi et al., 2015*; *Ramanaiah & Kadirvel, 2021*; *Ranathunge, Chimahusky & Welch, 2022*; *Ahmed et al., 2022*; *Dudhe et al., 2024*). SSRs, being highly polymorphic and codominant, are widely used in genome mapping, gene tagging, diversity estimation, variety identification, and marker-assisted selection (*Ozkan et al., 2022*). This study was aimed to assess SSR markers' performance in revealing the genetic diversity and population structure of sunflower genotypes, identifying suitable germplasms. It also explores using genetic divergence from morphological and molecular data to select elite germplasms, providing valuable information for sunflower breeding programs.

## MATERIAL AND METHODS

### Location and climatic conditions

The study was carried out at the Oilseeds Department, Centre for Plant Breeding and Genetics, Tamil Nadu Agricultural University, Coimbatore situated at an elevation of 426.72 m above mean sea level and located between 11°N latitude and 77°E longitude. The research station recorded an average annual rainfall of approximately 640 mm during *kharif* (rainy) season. The soil is primarily sandy clay loam, and the climate is semi-arid.

### Experimental material

Forty-eight sunflower germplasms, along with three check varieties (COSF 6B, CSFI 99, and COSFV 5), were included in the experiments. These germplasms, sourced from various agroclimatic zones from, Indian Institue of Oilseeds Research, Hyderabad (18); University of Agricultural Sciences, Bangalore (17); University of Agricultural Sciences, Raichur (5); Tamil Nadu Agriculture University, Coimbatore (4); Panjab Agricultural University, Ludhiana (3); and Regional Agricultural Research Stastion, Nandyal (1) are detailed in Table S1. This study aimed to examine the morphological and molecular diversity of these diverse germplasms from various locations, with the goal of identifying high-yielding and high-oil-content genotypes. The evaluation and characterization of these accessions took place during the *kharif* season of 2022, in an augmented block design I with three replication for check varieties for nine traits. Planting was done with (45x30) cm inter- and intra- row spacing. Standard agricultural practices were followed in accordance with the TNAU package of practices (*TNAU, 2024*) to ensure the cultivation of healthy crops.

### Morphological characterization

Data was collected from five plants chosen at random for nine traits from each accession, including days to 50% flowering (DF), days to maturity (DM), plant height (cm) (PH),

head diameter (cm) (HD), hundred seed weight (g) (HSW), volume weight per 100 ml (g) (VW), oil content (%) (OC), yield per plant (g) (SYP) and oil yield per plant (g) (OYP). Days to flowering were determined by counting the days from sowing to the opening of ray florets. Plant height (cm) was measured from ground level to the base of the capitulum a physiological maturity. Head diameter (cm) was recorded at physiological maturity. Volume weight (g/100 ml) was determined by filling seeds in a 100 ml measuring jar and weighing them, expressed in grams per 100 ml. Hundred-seed weight (g) was calculated by randomly selecting and weighing 100 seeds, expressed in grams. Seed yield per plant (g) was obtained by cleaning and weighing the harvested seeds from a single plant, expressed in grams per plant.

### Estimation of oil content (%)

The oil content (%) was determined using the Socsplus—SCS 08 AS apparatus. Sunflower kernels are dried, crushed, and 2–3 grams of crushed kernels were weighed and taken for analysis. A filter paper holds the sample in a thimble submerged in petroleum ether. The solvent is boiled at 80 °C for 1 h, followed by 140 °C for 30 min to evaporate it. The solvent drips through the thimble three times before stopping. The final weight of the beaker containing the extracted oil was recorded, and the solvent is collected in an empty beaker for reuse. Oil yield is expressed in grams per plant and calculated using the formula.

$$\text{Oil content (\%)} = \frac{\text{breker with oil weight} - \text{empty beaker weight}}{\text{kernal dry weight}} \times 100$$

$$\text{Oil yield per plant (g)} = \frac{\text{seed yield per plant} \times \text{oil content (\%)}}{100}.$$

### DNA extraction and marker amplification

The genomic DNA was extracted from young leaves using the CTAB (Cetyl Trimethyl ammonium bromide) method, following the procedure outlined by *Doyle & Doyle (1987)*. Subsequently, DNA quality was checked on a 0.8% Agarose gel. The concentration of DNA was determined using a nanoquant ND-1000 (Thermo Fisher Scientific, Waltham, MA, USA) and normalized to 10 ng/μl. In this study, publicly accessible ORS primers, as mapped by *Tang et al. (2002)*, were employed. A total of 103 SSR primers were used to evaluate the diversity among the germplasms. These markers were evenly distributed across the seventeen linkage groups (LGs) of sunflower, with an average of five to eight markers per LG, primers were used to assess molecular diversity.

For polymerase chain reaction (PCR), the reaction mixtures consisted of 2 μL of DNA, 3 μL of Master Mix (2X), 1 μL of 10 pMol primer, and 4 μL of nuclease-free water. PCR amplification was performed in a thermocycler (Applied Biosystems, Waltham, MA, USA) with the following conditions: an initial cycle at 94 °C for 3 min (denaturation), followed by a touch-down phase with the annealing temperature decreasing by 0.5 °C per cycle for 20 cycles (94 °C for 30 s, 63 °C for 30 s, 72 °C for 1 min). This was followed by 20 cycles of 94 °C for 15 s, 55 °C for 30 s, and 72 °C for 1 min, concluding with a final extension step at 72 °C for 10 min. The PCR products obtained were run on a 3% agarose gel and observed

using the GELSTAIN 49 advanced gel documentation unit (M/s Medicare, India). The molecular weight of the amplified bands was determined by comparing them with a 100 base pair ladder.

## Statistical analysis of molecular marker and quantitative traits

In this study, each SSR band was considered an independent locus and scored as a dominant marker, with each locus assigned a value of 1 for the presence of a band and 0 for its absence. The binary data (0/1) were used to compute the average polymorphism information content (PIC) and resolving power (Rp) following the methodology outlined by *Sharma et al. (2017)*. The multiplex ratio, effective multiplex ratio, and marker index were calculated according to the procedure described by *Powell et al. (1996)*. The allelic data were subsequently used to estimate genetic distances and to construct a neighbor-joining tree (*Saitou & Nei, 1987*) with the DARwin 6.0 software package (*Perrier & Jacquemoud-Collet, 2006*).

GenAlEx V6.5 was employed for Analysis of Molecular Variance (AMOVA), Principle Coordinate Analysis (PCoA), and other parameters including number of observed alleles (na), number of effective alleles (ne), Nei's gene diversity (h), Shannon's information index (I) and gene flow (Nm) for analyzing inter and intra-population diversities (*Gogoi et al., 2023*).

To investigate population structure, the model-based program STRUCTURE 2.3.4 (*Pritchard et al., 2000*) was utilized, with three replications conducted for each K value. Each run included a burn-in period of 100,000 steps followed by 100,000 Monte Carlo Markov Chain replicates. The membership of each genotype was assessed across a range of genetic clusters from $K = 1$ to 20, using the admixture model and assuming correlated allele frequencies. The plateau of $\Delta$K was detected by graphing LnPD values for each K. The ultimate population structure was established using the "StructureSelector" software (*Evanno, Regnaut & Goudet, 2005*; *Puechmaille, 2016*; *Li & Liu, 2018*).

To evaluate morphological diversity among germplasms, we utilized the 'prcomp' function for Principal Component Analysis (PCA). Subsequently, the 'biplot' function was employed to visualize PC1 *versus* PC2. For hierarchical cluster analysis, we leveraged the 'hclust' function within the R Studio environment to conduct statistical data analysis.

## RESULTS

### Analysis of SSR markers and molecular clustering patterns

The initial assessment screened 130 SSR primers chosen from literature review. Out of these, 103 primers demonstrated efficient traits like robust amplification, reproducible patterns, and multiple polymorphic fragments per assay, while the rest were monomorphic. Detailed outcomes including polymorphism percentages, polymorphism information content (PIC), resolving power (Rp), and marker index (MI) are provided in the accompanying Table S2. The screened primers produced well-defined amplified bands, with allele counts ranging from 2 to 6 per individual across all loci. A total of 267 alleles were identified, averaging 2.6 alleles per locus, with ORS621 showing the highest count at 6 alleles. All alleles exhibited polymorphism, resulting in a 100% polymorphism percentage, a notable finding in this
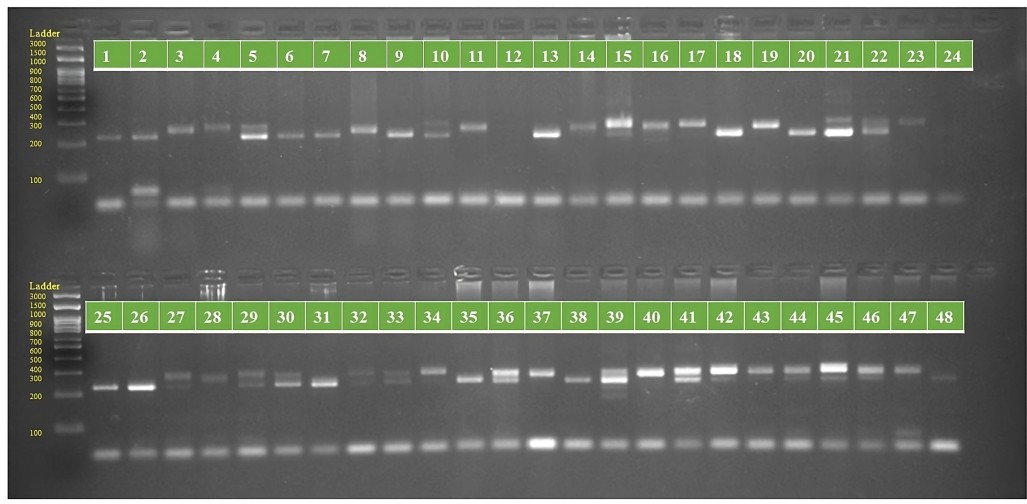

**Figure 1  PCR amplification profiles of SSR marker ORS 807 for 48 sunflower genotypes.** The genotypes are labeled as follows: 1. ARM 248B, 2. COSF 6B ©, 3. GMU 1181, 4. RHA 102, 5. CMS 1103B, 6. GMU 780, 7. CMS 335B, 8. RCR 72, 9. CMSNDCMS2B, 10. GMU 344, 11. CSFI 99 ©, 12. RHA 272-1, 13. GMU 336, 14. RHA GPR 58, 15. CMS 911B, 16. GMU 477, 17. GMU 450, 18. PM 95, 19. GMU 428, 20. GMU 411, 21. GMU 741, 22. CMS 108B, 23. COSFV ©5, 24. GMU 734, 25. ARM 240B, 26. CMS 597B, 27. HOCL 6R, 28. RHA GPR 110, 29. IB 80, 30. CMS 107B, 31. PM 36, 32. IL 77, 33. RHA95-C-10, 34. IL84, 35. RHA 278, 36. RHA 273, 37. RHA 857, 38. GMU 755, 39. PM 53, 40. GP6 912, 41. CMS 135B, 42. REC 431, 43. GMU 325, 44. GP6 1089, 45. RHA 378, 46. PM 65, 47. RHA GMU 755 and 48. COSF 13B.

study. PCR amplification of SSR markers for 48 sunflower genotypes are shown in Figs. S1–S6 and Figs. 1–2. The average PIC value among all primers ranged from 0.08 to 0.49 in this survey. Primer ORS853 exhibited the minimum average PIC value, whereas primer ORS613 displayed the maximum. Primer Index values fluctuated from 0.16 to 1.94, with ORS725 having the highest and ORS853 the lowest values. Resolving Power (Rp) values varied from 0.17 to 5.67, with the highest observed in ORS887 and the lowest in ORS1120. The mean Rp values ranged from 0.08 to 1.77, with ORS565 having the highest and ORS1120 the lowest. The average PIC was calculated to be 0.30, while the Multiplex Ratio (MR), Effective Multiplex Ratio (EMR), and Marker Index (MI) were determined to be 2.59, 2.59, and 0.78, respectively.

Cluster analysis through neighbor joining method with jaccard's pairwise distance matrix, based on SSR marker data from 48 sunflower germplasm accessions revealed five distinct clusters. Cluster I comprised 12 genotypes, cluster II had 7 genotypes, cluster III contained 14 genotypes, cluster IV included 5 genotypes and cluster V had 10 genotypes (Fig. 3 and Table S3). The jaccard's pairwise coefficient indicated a maximum genetic dissimilarity of 0.65 among accessionRHA 272-1 and COSF 13B; RHA95-C-10 and COSF 13B. Conversely, there was a minimum genetic dissimilarity of 0.33 concerning HOCL 6R and CMS 107B (Table S4).

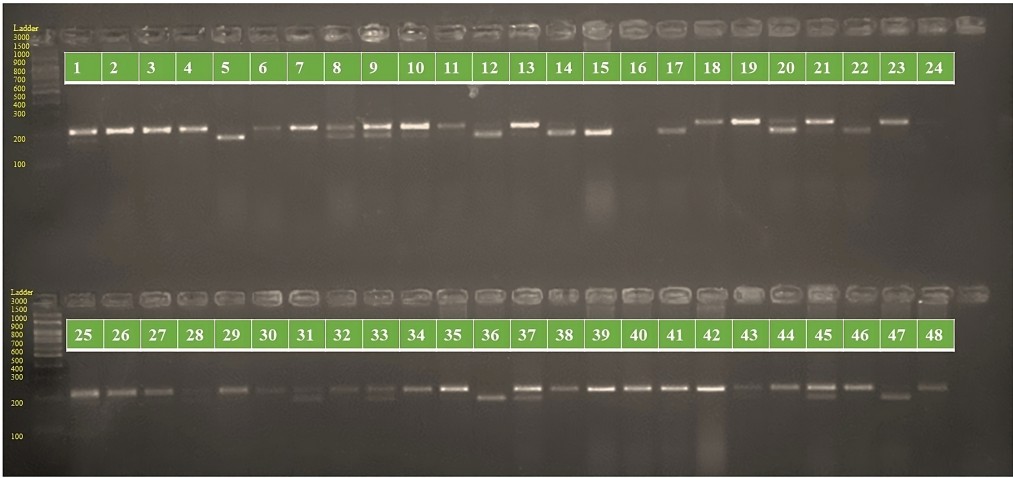

**Figure 2  PCR amplification profiles of SSR marker ORS 1271 for 48 sunflower genotypes.** The genotypes are labeled as follows: 1. ARM 248B, 2. COSF 6B ©, 3. GMU 1181, 4. RHA 102, 5. CMS 1103B, 6. GMU 780, 7. CMS 335B, 8. RCR 72, 9. CMSNDCMS2B, 10. GMU 344, 11. CSFI 99 ©, 12. RHA 272-1, 13. GMU 336, 14. RHA GPR 58, 15. CMS 911B, 16. GMU 477, 17. GMU 450, 18. PM 95, 19. GMU 428, 20. GMU 411, 21. GMU 741, 22. CMS 108B, 23. COSFV ©5, 24. GMU 734, 25. ARM 240B, 26. CMS 597B, 27. HOCL 6R, 28. RHA GPR 110, 29. IB 80, 30. CMS 107B, 31. PM 36, 32. IL 77, 33. RHA95-C-10, 34. IL84, 35. RHA 278, 36. RHA 273, 37. RHA 857, 38. GMU 755, 39. PM 53, 40. GP6 912, 41. CMS 135B, 42. REC 431, 43. GMU 325, 44. GP6 1089, 45. RHA 378, 46. PM 65, 47. RHA GMU 755 and 48. COSF 13B.

## Principal Coordinate Analysis

Principal coordinate analysis (PCoA) emerges as a robust statistical technique leveraging the dissimilarity matrix derived from allelic data. It facilitates the estimation of individual and group disparities, detection of outliers, and validation of genetic relationships among genotypes. The resulting two-dimensional PCoA plot closely resembles the groupings obtained through the Neighbor Joining method using Jaccard's pairwise distance matrix in cluster analysis. The PCoA plot distinctly delineates genotypic differences originating from different geographical locations. Notably, the analysis indicates that the first three axes (PC1, PC2, and PC3) cumulatively account for 27.24% of the total variation, with individual contributions of 10.50%, 9.69%, and 7.04%, respectively (Table 1). Among the genotypes examined, RCR 72, GMU 325, PM 36, RHA 272-1, CMS 335B, CMS 1103B, CMS 911B, RHA 273, RHA857, COSF 13B, RHA GMU 755, GMU 734, and HOCL 6R are notably distant from the cluster centroid, while the remaining varieties closely cluster around it (Fig. 4).

## Analysis of Molecular Variance (AMOVA)

Molecular variance within sunflower populations was evaluated through Analysis of Molecular Variance (AMOVA). The results revealed a notable predominance of variation within populations, accounting for 89%, whereas variation among populations constituted a significantly lower proportion of 11% (Table 2 and Fig. 5).

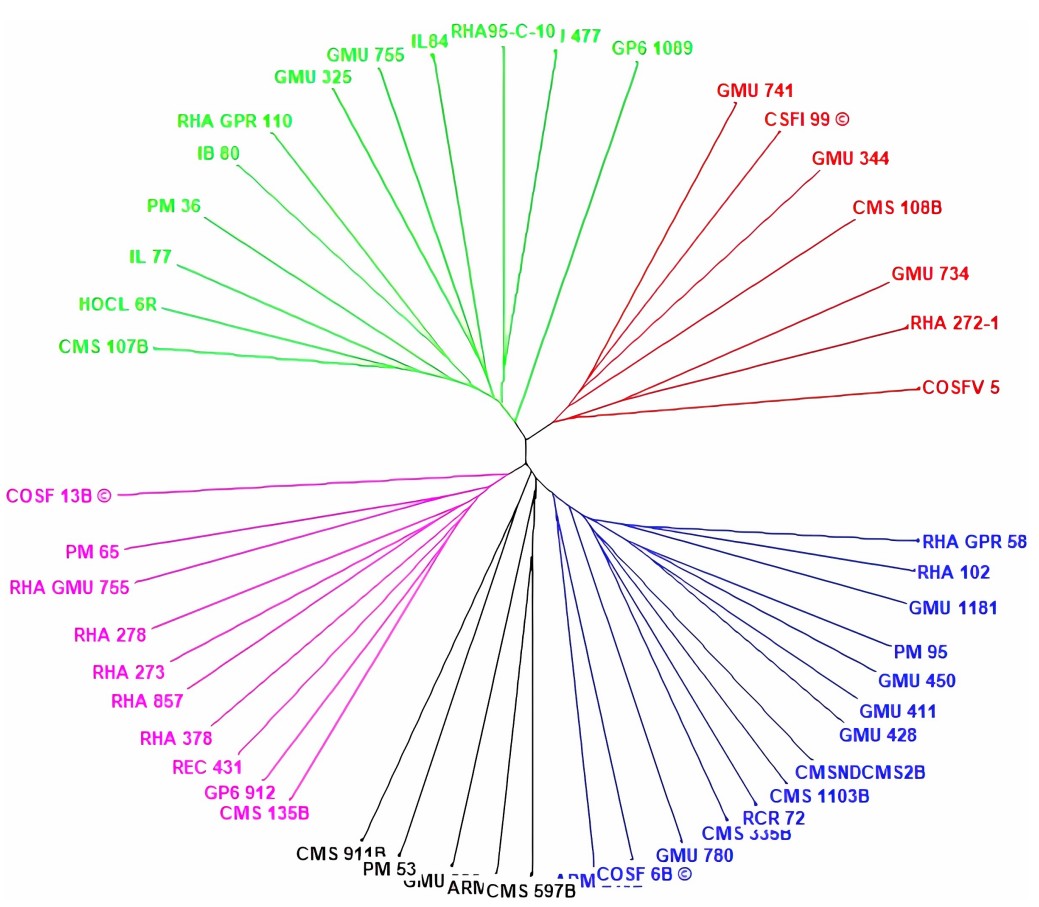

**Figure 3 Constructing a neighbor-joining tree of 48 genotypes utilizing DARwin software and data from 103 SSR markers.** The resulting dendrogram revealed five distinct clusters: Cluster I (green) consisted of 12 genotypes; Cluster II (red) had 7 genotypes; Cluster III (blue) contained 14 genotypes; Cluster IV (black) included five genotypes; and Cluster V (pink) comprised 10 genotypes.

**Table 1 Percentage of variation explained by the first 3 axes of Principal Coordinate Analysis (PCoA).**

| S. no | Axis | 1 | 2 | 3 |
|-------|------|-----|-----|-----|
| 1 | Percent of variation | 10.50 | 9.69 | 7.04 |
| 2 | Cumulative percent of variation | 10.50 | 20.20 | 27.24 |

## Model based population structure

The structure analysis based on SSR marker data grouped the 48 sunflower germplasm into five distinct subpopulations. Utilizing Bayesian methods in structure software facilitated efficient population structure analysis, assigning individuals to appropriate subpopulations. Admixture model-based simulations determined the optimal number of subpopulations ($K = 5$), confirming the subdivision (Fig. 6). Genetic relatedness among populations was assessed using structure, with membership coefficients indicating subpopulation affiliation. Accessions scoring above 0.80 were considered genetically pure, while those below were categorized as intermixed. Similarly, 48 sunflower genotypes were divided into

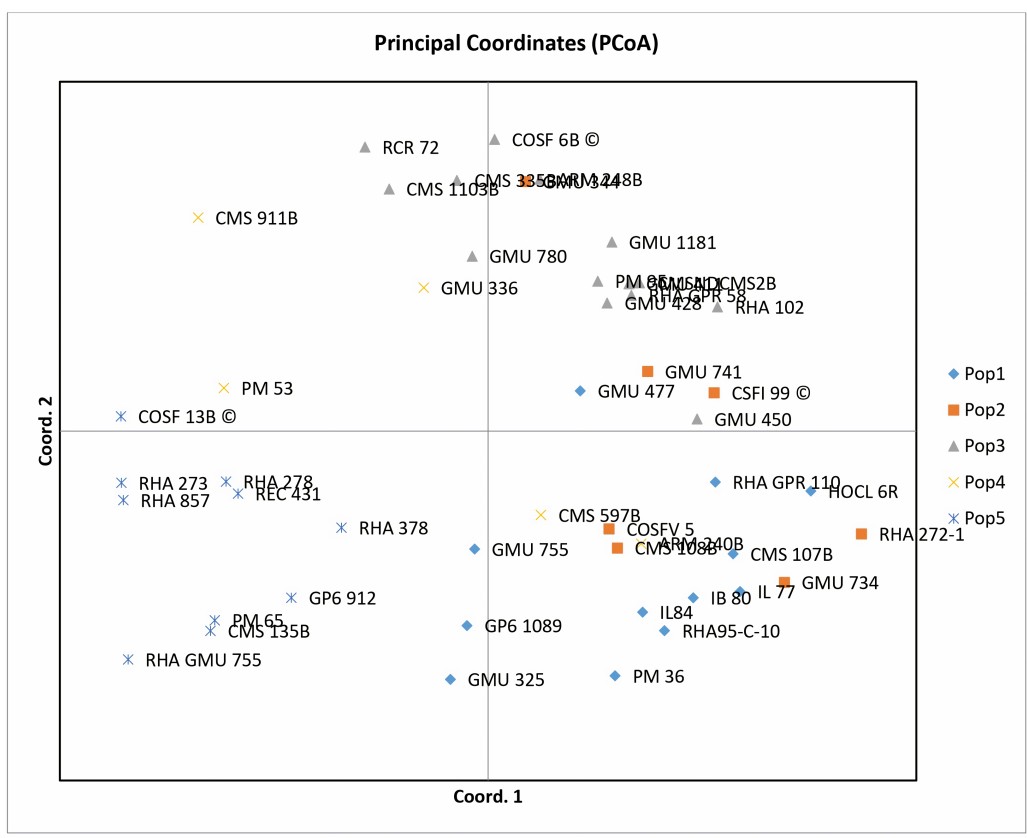

**Figure 4** Performing principal coordinate analysis on sunflower genotypes using SSR markers data.

five subpopulations, with the following representation percentages: 20.83% (subpopulation I, 10 genotypes), 22.91% (subpopulation II, 11 genotypes), 11.25% (subpopulation III, 9 genotypes), 12.50% (subpopulation IV, 6 genotypes), and 25.00% (subpopulation V, 12 genotypes) represented in Fig. 7. Net nucleotide distances among subpopulations revealed varying degrees of genetic divergence, with subpopulations II and V exhibiting the highest distance (0.056) and subpopulations I and IV showing the lowest (0.014). This diversity was further emphasized by expected heterozygosity variations: subpopulation IV showcased the highest diversity (0.352), while subpopulation III displayed the lowest, accompanied by a low Fst value (0.173). Subpopulation II showed the highest Fst mean percentage and the lowest expected heterozygosity (0.2277), while subpopulation IV had the lowest Fst mean percentage and the highest expected heterozygosity. Significant divergence (Fst) is evident among sub-populations represented in (Fig. 8 and Table 3) and plotting the structure of significant divergence among sub-populations triangular plot, histogram distribution of alpha and histogram distribution of Fst presented in (Fig. 9 and Figs. S7–S8).

## Genetic diversity within and among populations

The genotypes were divided into five populations, and various genetic variability parameters such as the number of observed alleles (na), Nei's gene diversity (h), number of effective

**Table 2** Summary of molecular variance within sunflower populations was evaluated through Analysis of Molecular Variance (AMOVA).

| Source | df | SS | MS | Est. var. | % | *P*-value |
|---|---|---|---|---|---|---|
| Among Pops | 4 | 326.917 | 81.729 | 4.711 | 11% | 0.001[**] |
| Within Pops | 43 | 1625.979 | 37.813 | 37.813 | 89% | 0.001[**] |
| Total | 47 | 1952.896 | 119.542 | 42.524 | 100% | |

**Notes.**
df, Degree of Freedom; SS, Sum Square; MS, Mean sum square; Est. Var, Estimated variance.
Two asterisks (**) indicate significance at the 1% probability level.

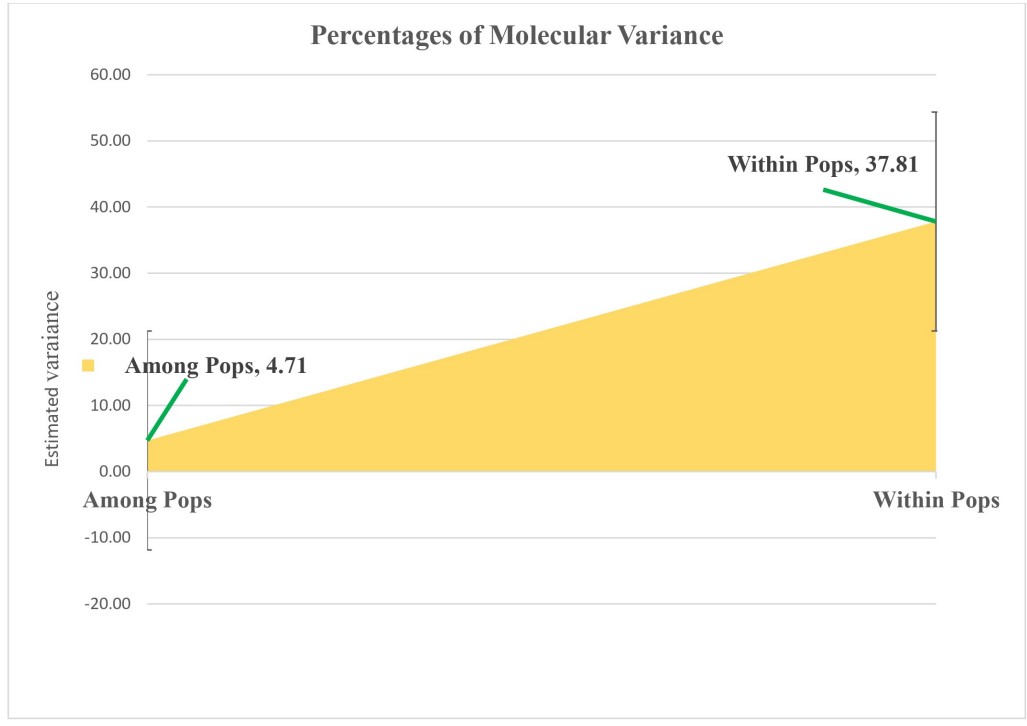

**Figure 5** Analyzing molecular variance within and among populations of sunflower using AMOVA.
The estimated variation within populations accounted for 37.81, while the variation among populations represented a significantly lower proportion of 4.71. These values are plotted on the vertical axis.

alleles (ne), Shannon's information index (I), Nei's unbiased diversity (uh), and gene flow (Nm) were computed (Table 4). Among these parameters (na, ne, I, h, uh), subpopulation IV exhibited the highest values (1.74, 1.48, 0.43, 0.29, and 0.31), followed by Population I (1.54, 1.43, 0.38, 0.25, and 0.30), while Population II displayed the lowest values (1.24, 1.34, 0.29, 0.20, and 0.25). Furthermore, the gene flow (Nm) value was 4.01. The polymorphic percentage was highest in subpopulation IV, recording 83.52%, and lowest in subpopulation II at 51.31%. The allelic pattern across the population is illustrated in the Fig. 10, indicating that subpopulation III has a higher number of alleles distributed across the population, whereas subpopulation IV displays lower allelic distribution. Pairwise population matrices
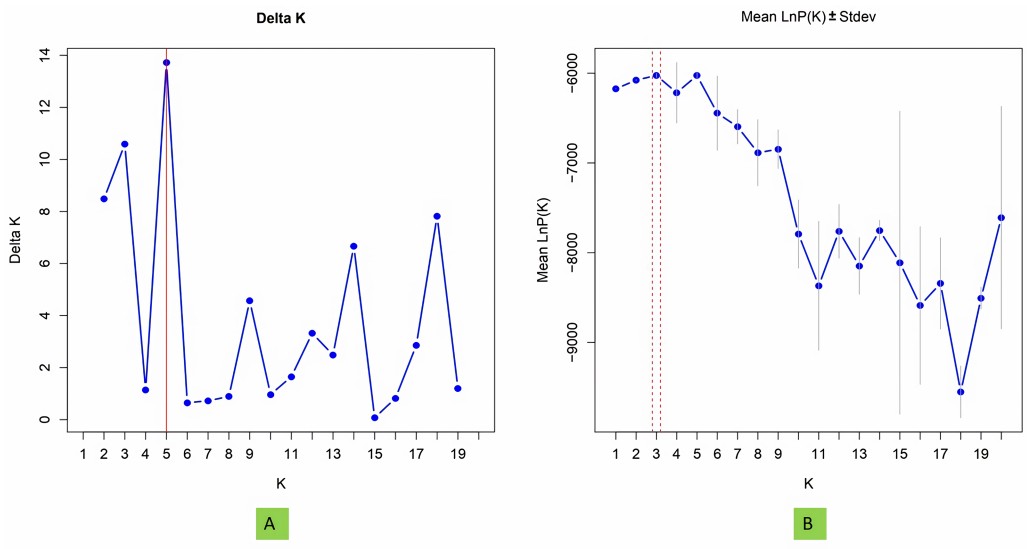

**Figure 6** **Estimation of hypothetical sub-populations using ΔK-values, (A) Delta K, (B) Mean LnP(K).** The optimal K value was determined to be 5, indicating the presence of five subpopulations.

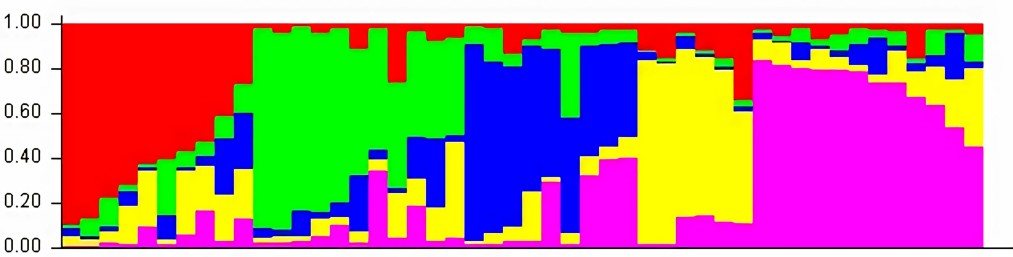

**Figure 7** **A model-based population structure plot for 48 accessions was generated using the STRUCTURE software with K = 5 and 103 polymorphic SSR markers.** The color scheme is as follows: sub-population I (10 genotypes) is represented in red, sub-population II (11 genotypes) in green, sub-population III (9 genotypes) in blue, sub-population IV (6 genotypes) in yellow, and sub-population V (12 genotypes) in pink. Each vertical line represents an individual genotype, with distinct segments indicating the degree of admixture within each genotype.

of Nei's unbiased genetic distance and Nei's unbiased genetic identity across the five populations were computed and analysed (Tables S5–S6).

## Genetic divergence based on morphological data

Principal Component Analysis (PCA) and cluster analysis are powerful multivariate techniques utilized to explore genetic diversity, trace crop evolutionary pathways, select parental lines, and assess diversity. PCA was employed to assess the relationships between various parameters. Nine principal components (PCs), corresponding to the number of traits, were identified through PCA (Table 5). With four PCs exhibiting eigenvalues greater than 1 as shown in (Fig. 11). Among these PC1 explained 36.67% of the variation,
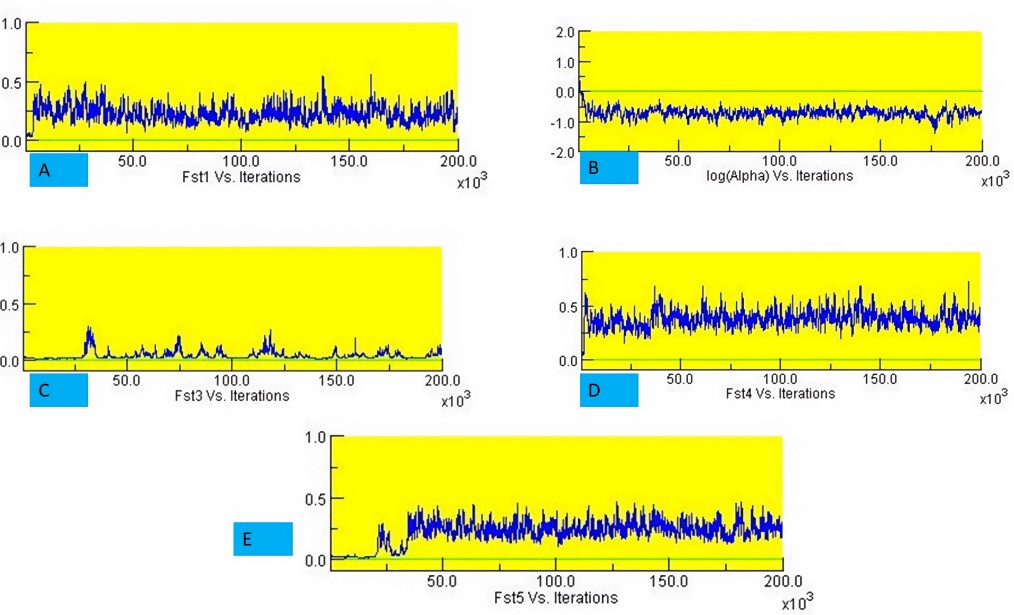

**Figure 8 Significant divergence (Fst) is evident among sub-populations.** (A) Fst1 *vs* Iterations, (B) Fst2 *vs* Iterations, (C) Fst3 *vs* Iterations, (D) Fst4 *vs* Iterations and (E) Fst5 *vs* Iterations.

followed by PC2 (24.04%), PC3 (14.67%), and PC4 (11.34%). Traits positive factor loading (PFL) with PC1 included VW, while PC2 exhibited PFL with DF, DM, PH, HSW, VW, and OC. Similarly, PC3 showed PFL with VW, OC, and OYP, while PC4 exhibited PFL with DF, DM, and SYP. The biplot of PC1 *vs.* PC2 visually depicts trait variability, with genotypes positioned towards the positive or negative ends of principal components reflecting favourable traits. DF, DM, HD, OYP, SYP and HSW these trait contribution to divergence of sunflower represented in (Fig. 12).The biplot also illustrates overlapping vectors for traits such as HD, OYP, PH, OC, HSW, OC, DM, DF, and SYP, indicating strong association among them. Genotypes such as RHA 273, COSFV5, RHA102, IB 80, GMU 325, and GP6 1089 are positioned away from the centroid, while the remaining genotypes are clustered closer to the centroid (Fig. 13). Hierarchical Cluster Analysis (HCA) grouped the 48 sunflower genotypes, including checks, into three clusters using PCs, eigenvalues surpassing 1, and principal factor scores (PF scores) for various traits. The analysis unveiled that cluster I is comprised of 19 genotypes, cluster II encompassed 15, and cluster III included 14 genotypes. Cluster I displayed the highest diversity among genotypes, while those in cluster III, exhibiting the lowest diversity (Fig. 14 and Table S7). Mean performance for various traits was observed based on morphological data early flowering and early maturity were noted in RHA 102, IB 80, CMS 107B, PM 36, GP6 912, and RHA 378. High head diameter was recorded in RHA 102, CMS 1103B, RHA GPR 58, COSF 13B, GMU 336, and GMU 1181. High oil content was observed in CMSNDCMS2B, GMU 477, GMU 450, PM 36, RHA 378, and COSF 13B. High seed yield per plant was

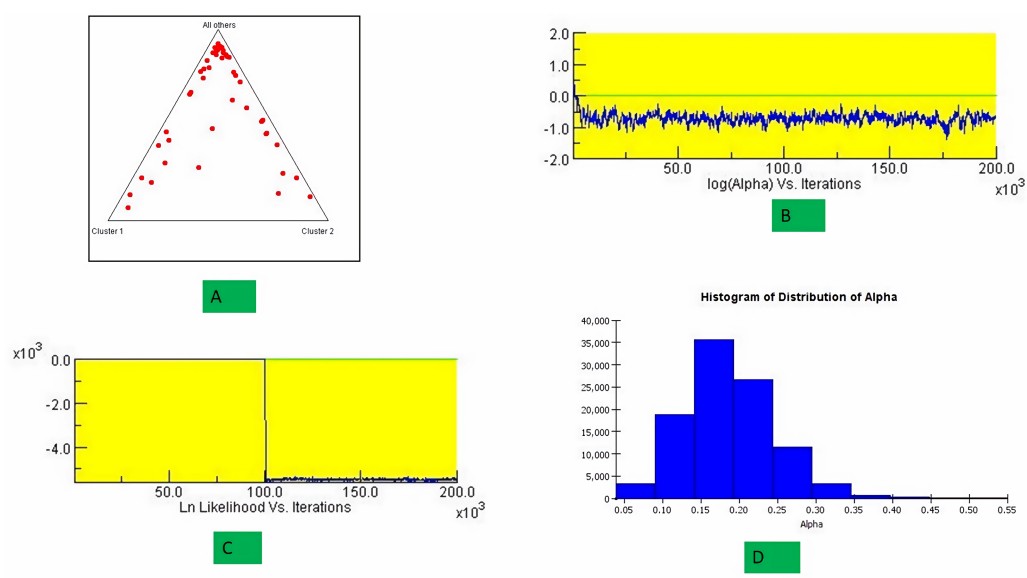

**Figure 9** **Plotting the structure of significant divergence among sub-populations.** (A) Triangular plot, (B) log (Alpha) *Vs* Iterations, (C) Ln Likelihood *Vs* Iterations (D) Histogram distribution of Alpha.

found in ARM 248B, GMU 1181, RHA 102, CMS 1103B, GMU 336, RHA GPR 58, COSF 13B, GMU 477, and GMU 450.

# DISCUSSION

Molecular markers are versatile tools for studying genetic diversity among plant populations. Traditionally, germplasms are selected based on biometrical data, which is time-consuming and less reliable due to environmental influences. Molecular markers, such as microsatellites (SSR), offer a more efficient and reliable alternative. SSRs, with their multiallelic and polymorphic nature, allow for the exploration of dissimilarity in parental lines of breeding material. Their stability, unaffected by environmental effects, provides a major gain over conventional methods using DNA markers. The number and size of DNA fragments depended on primer sequences, with reactions repeated for consistency. Only well-defined and bright DNA bands were counted. These results highlight SSR markers' utility in sunflower parent identification, superior genotypes selection. Traditional methods like grow-out trials are time-consuming and inclined by environmental factors. Molecular markers offer a faster and more reliable substitute for varietal identification, hybrid confirmation, and parental selection in sunflower breeding programs. Genetic impurity in cross-pollinated crops like sunflower poses a widespread challenge, and relying solely on phenotype-based descriptions is cumbersome and less dependable (*Iqbal et al., 2011*; *Zeinalzadeh-Tabrizi et al., 2018*; *Ahmed et al., 2022*).

Studying genetic variance is crucial for breeders seeking to pinpoint promising genotypes. In this investigation, SSR DNA markers were employed to scrutinize genetic connections among sunflower genotypes. The prevalence of alleles per locus indicates

Peer*J*

**Table 3** Substantial divergence (Fst) observed between sub-populations, along with average distances (expected heterozygosity) among sunflower populations, and net nucleotide distances among sub-populations.

| Sub population | Fst value | Percent mean value of Fst within population | Genotypes | Exected heterozygosity | Percentage of genotypes | Net nucleotide distance | | | | |
|---|---|---|---|---|---|---|---|---|---|---|
| | | | | | | Sub-population I | Sub-population II | Sub-population III | Sub-population IV | Sub-population V |
| I | 0.183 | 0.2109 | 10 | 0.3019 | 20.83 | – | 0.0539 | 0.049 | 0.0141 | 0.045 |
| II | 0.202 | 0.388 | 11 | 0.2277 | 22.91 | 0.0539 | – | 0.0474 | 0.0339 | 0.0557 |
| III | 0.173 | 0.3292 | 9 | 0.2655 | 18.75 | 0.049 | 0.0474 | – | 0.0254 | 0.0467 |
| IV | 0.188 | 0.0004 | 6 | 0.352 | 12.5 | 0.0141 | 0.0339 | 0.0254 | – | 0.0179 |
| V | 0.254 | 0.2275 | 12 | 0.2816 | 25 | 0.045 | 0.0557 | 0.0467 | 0.0179 | – |

**Table 4  Genetic diversity of sunflower population from using following parameters (number of observed alleles (na), number of effective alleles (ne), Nei's gene diversity (h), Shannon's information index (I), Nei's gene diversity (h) and unbiased diversity (uh) and gene flow (Nm).**

| Population | Mean | | | | | Polymorphic % | Nm |
|---|---|---|---|---|---|---|---|
| | Na | Ne | I | h | uh | | |
| 1 | 1.54 | 1.43 | 0.38 | 0.25 | 0.30 | 69.29% | |
| 2 | 1.24 | 1.34 | 0.29 | 0.20 | 0.25 | 51.31% | |
| 3 | 1.42 | 1.40 | 0.35 | 0.23 | 0.28 | 62.55% | |
| 4 | 1.74 | 1.48 | 0.43 | 0.29 | 0.31 | 83.52% | 4.01 |
| 5 | 1.25 | 1.37 | 0.32 | 0.21 | 0.27 | 54.68% | |
| Total | 1.44 | 1.40 | 0.35 | 0.24 | 0.28 | 64.27% | |
| SE | 0.05 | 0.02 | 0.02 | 0.01 | 0.01 | | |

**Notes.**

Na, No. of Different Alleles; Ne, No. of Effective Alleles; I, Shannon's Information Index; h, Diversity; uh, Unbiased Diversity.

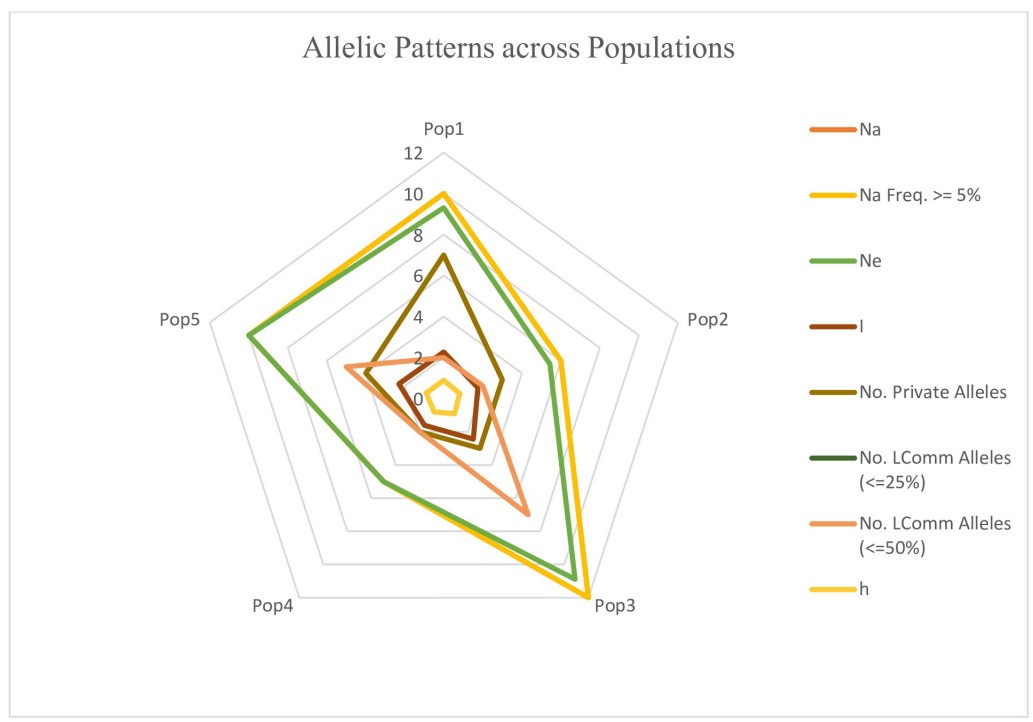

**Figure 10  Distribution of allelic patterns among five populations of sunflower genotypes using molecular data.** The allelic patterns among subpopulations are represented by different colors: yellow indicates the number of different alleles (Na, frequency >= 5%); green represents the number of effective alleles (Ne); light brown shows the number of private alleles; light pink indicates the number of different alleles (Na); brown shows Shannon's Information Index (I); and light yellow represents genetic diversity (h).

natural outcrossing in sunflowers, a crop that relies on cross-pollination. Primer Index (PI), Resolving Power (Rp), Effective Multiplex Ratio (EMR), and Marker Index (MI) evaluate the efficiency of molecular markers in genetics, these metrics help to select effective

**Table 5  Summary of principal component analysis and eigenvalues for diverse traits observed in sunflower genotypes.**

| S.no | PC1 | PC2 | PC3 | PC4 | PC5 | PC6 | PC7 | PC8 | PC9 |
|------|------|------|------|------|------|------|------|------|------|
| DF | −0.241 | 0.571 | −0.058 | 0.283 | 0.012 | −0.174 | 0.017 | 0.569 | 0.422 |
| DM | −0.257 | 0.560 | −0.025 | 0.287 | 0.002 | −0.186 | 0.093 | −0.566 | −0.418 |
| PH | −0.241 | 0.074 | −0.620 | −0.177 | 0.557 | 0.361 | −0.280 | −0.037 | −0.008 |
| HD | −0.461 | −0.294 | −0.117 | −0.009 | 0.132 | −0.017 | 0.817 | 0.050 | 0.015 |
| HS | −0.337 | 0.160 | −0.162 | −0.361 | −0.738 | 0.398 | −0.033 | 0.004 | −0.009 |
| VW | 0.059 | 0.293 | 0.099 | −0.817 | 0.174 | −0.437 | 0.106 | 0.005 | 0.002 |
| SYP | −0.459 | −0.305 | −0.005 | 0.017 | −0.064 | −0.417 | −0.381 | 0.355 | −0.496 |
| OC | −0.205 | 0.163 | 0.686 | −0.084 | 0.301 | 0.514 | −0.005 | 0.196 | −0.248 |
| OYP | −0.485 | −0.193 | 0.302 | −0.025 | 0.059 | −0.130 | −0.297 | −0.433 | 0.583 |
| PV | 36.670 | 24.040 | 14.670 | 11.340 | 6.704 | 5.106 | 1.368 | 0.061 | 0.041 |
| CV | 36.670 | 60.710 | 75.380 | 86.720 | 93.424 | 98.530 | 99.897 | 99.959 | 100.000 |
| EIGEN | 3.300 | 2.164 | 1.320 | 1.021 | 0.603 | 0.460 | 0.123 | 0.006 | 0.004 |
| SD | 1.817 | 1.471 | 1.149 | 1.010 | 0.777 | 0.678 | 0.351 | 0.074 | 0.061 |
| %V | 36.67 | 24.04 | 14.67 | 11.34 | 6.704 | 5.106 | 1.368 | 0.061 | 0.041 |
| CP | 36.67 | 60.71 | 75.38 | 86.72 | 93.424 | 98.53 | 99.897 | 99.959 | 100 |

**Notes.**

DF, Days to 50% flowering; DM, Days to Maturity; PH, Plant height; HD, Head diameter; HS, Hundred Seed weight; VW, Volume weight per 100 ml; OC, Oil content; SYP, Yield per plant; OYP, Oil yield per plant; SD, standard deviation; %V, Percent of variation; CP, Cumulative variation.

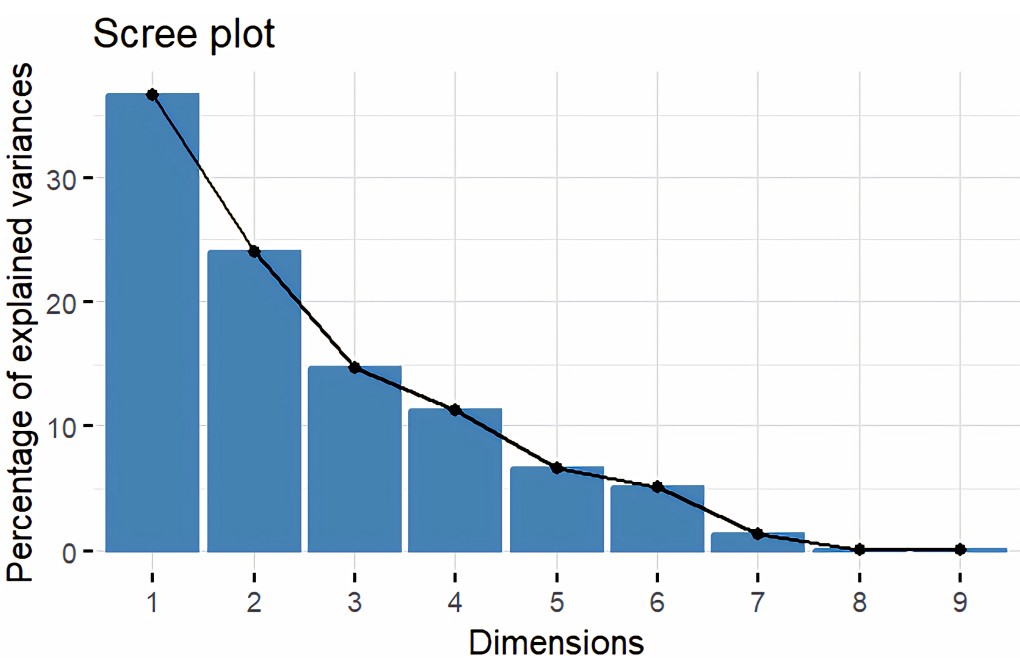

**Figure 11  Screeplot showing eigen value variance.**

markers and primer sets, optimizing genetic analysis. However, not all markers are equally effective for genotyping and varietal identification; their utility primarily hinges on their PIC value, Rp, and MI values, which signify their efficacy for genotypic differentiation (*Lavudya et al., 2024*). *Ibrar et al. (2022)* reported a range of 1 to 4 alleles, with an average of 2.1 alleles. Similarly, *Antonova et al. (2006)* employed 10 SSR primer pairs to assess the genetic diversity of 17 sunflower hybrids and inbred lines, revealing an average of 2.2 alleles. Most robust markers with high average PIC value indicates a highly informative marker that can clearly distinguish between individuals and populations, making it crucial for selecting markers in genetic studies. It's recommended to steer clear of markers with low PIC values when conducting genomic studies, favoring those with high or moderate values instead. *Ibrar et al. (2022)* pinpointed markers like ORS-605 and ORS-700 as fitting for assessing genetic diversity among germplasms. Similarly, *Sahranavard Azartamar et al. (2015)* emphasized markers HA3040 and ORS-733, noted for their high PIC values, as adept for analyzing genetic diversity in sunflowers. The most potent marker, ORS-453, was singled out by *Ahmed et al. (2022)*. While diversity profiling through SSR primers indicated a moderate level of genotypic diversity, the PIC values closely mirrored those reported by *Lochner (2011)* (ranging from 0.06 to 0.75) and *Darvishzadeh et al. (2010)* (ranging from 0.09 to 0.62).

Genotypes were classified using DARWIN v.6.0 software, cluster III exhibits a greater number of genotypes, suggesting higher genetic diversity within that cluster. Conversely, Cluster IV displays fewer genotypes, indicating lower genetic diversity, and implying that the genotypes within it are more similar to each other. The maximum jaccard's pairwise

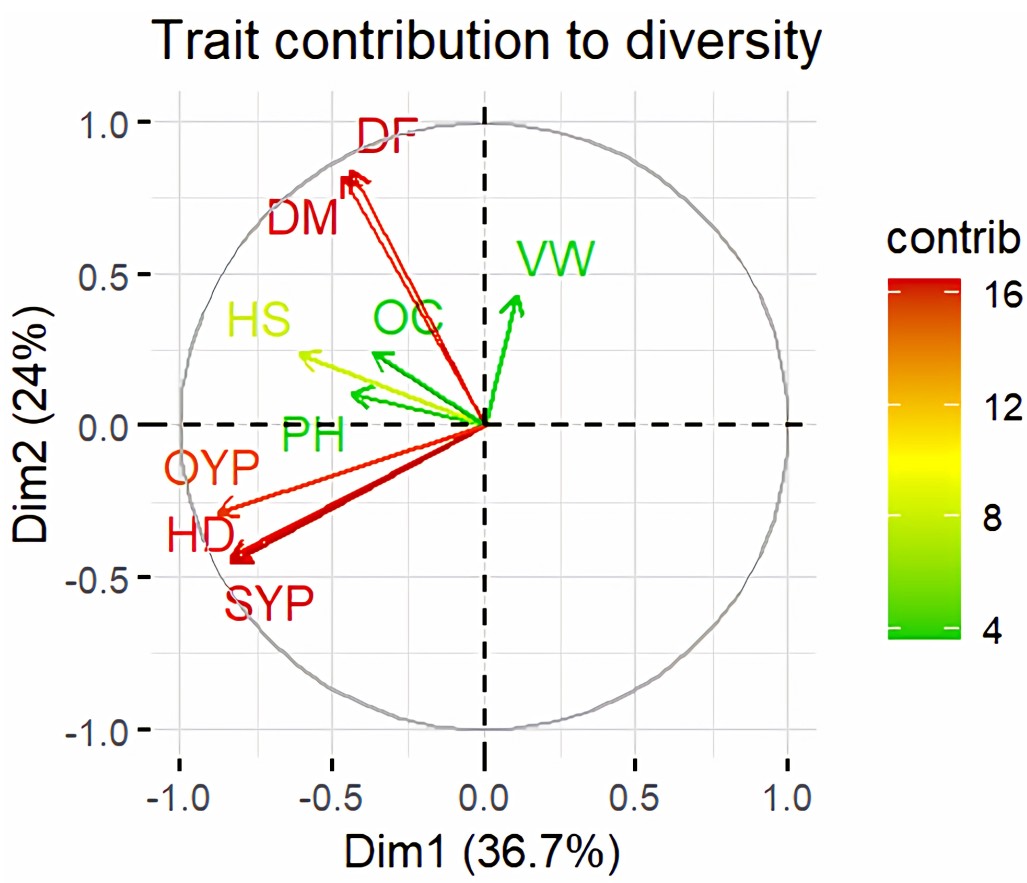

**Figure 12 Trait contribution to divergence of sunflower genotypes.** The traits are showed by color : red colour traits include DF (Days to 50% flowering), DM (Days to Maturity), HD (Head diameter), SYP (Seed yield per plant), and OYP (Oil yield per plant); dark green colour traits include PH (Plant height), VW (Volume weight per 100 ml), and OC (Oil content); light green colour trait is HS (Hundred seed weight).

coefficient revealed a determined genetic dissimilarity, indicating significant variation between genotypes. Conversely, there was a minimum genetic dissimilarity, suggesting significantly less variation between genotypes. *Ahmed et al. (2022)* conducted a study where the dendrogram analysis distinctly separated maternal and paternal genotypes, as well as their hybrids, into three primary clusters (A, B & C). Molecular characterization using SSR divided the sunflower lines into two main groups of Restorer and CMS lines, consistent with the results of morphological characterization (*Ibrar et al., 2022*). Similarly, *Pandey et al. (2018)* classified thirty-one genotypes into three major clusters. *Jannatdoust et al. (2016)* studied the hierarchical clustering of individuals using the Neighbor Joining method in DARwin5 software, resulting in the subdivision of individuals into three groups.

The resulting two-dimensional PCoA plot truly captures the groupings observed in the Neighbor Joining method, employing Jaccard's pairwise distance matrix for cluster analysis, suggests that these axes are important for understanding the underlying patterns or structure within the data, although there is still a considerable amount of variation not

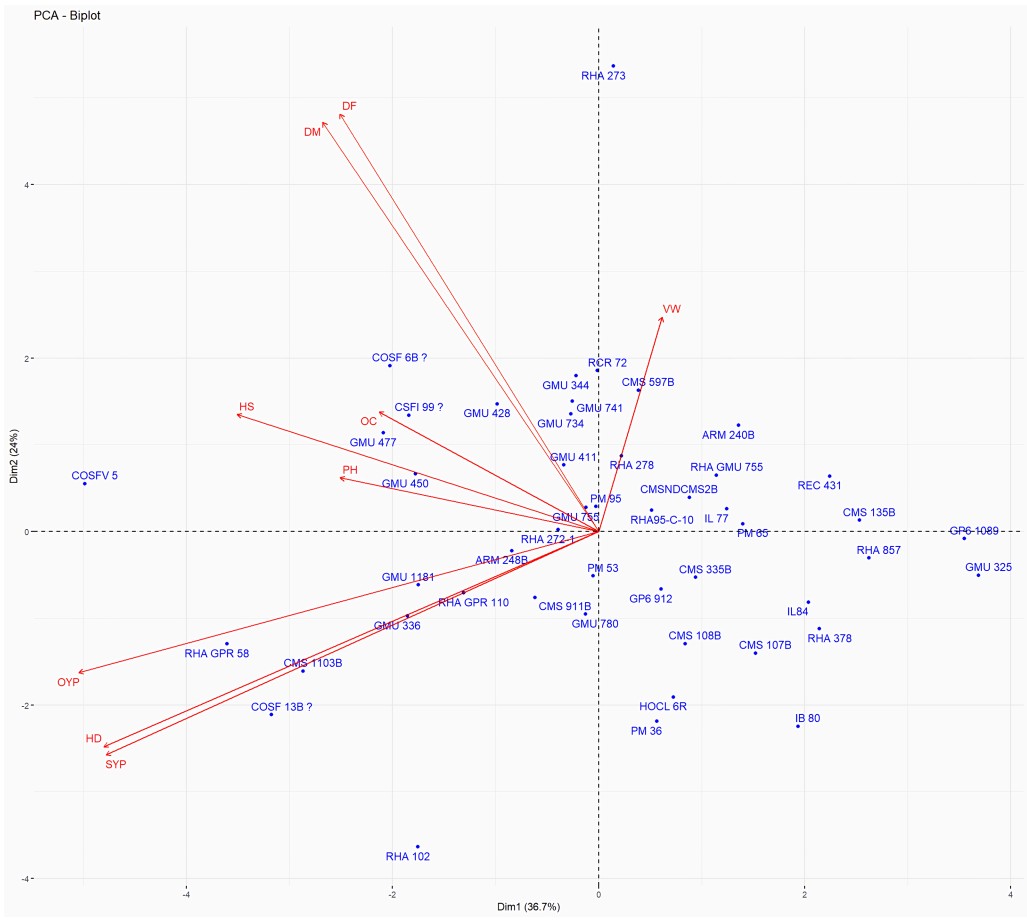

**Figure 13** **Forty eight genotypes of sunflower PC1 and PC2 bi-plots.** The traits are as follows: DF, Days to 50% flowering; DM, Days to Maturity; PH, Plant height; HD, Head diameter; HS, Hundred Seed weight; VW, Volume weight per 100 ml; OC, Oil content; SYP, Seed yield per plant; OYP, Oil yield per plant.



captured by these three axes alone. In PCoA, genotypes far from the cluster centroid suggest significant genetic differences from the cluster's average profile, potentially indicating outliers or distinct genetic traits within the population, genotypes close to the cluster centroid reflect genetic profiles that closely match the average characteristics of the cluster, implying population homogeneity or similarity. In a study by *Perveen et al. (2023)*, the first two principal axes contribute 18.12 and 12.31percent to the total genetic variation, separately, resulting in a cumulative variation of 30.43%. *Hladni et al. (2018)* demonstrate that principal coordinates provide insight into 58.4% of the total variability. The utility of PCoA in genetic diversity analysis is emphasized by *Jannatdoust et al. (2016)*, where the first and second components account for 7.86% and 6.16% of the total variance, respectively.

AMOVA analyzed molecular variance within sunflower populations, revealing substantial within-population variation, largely influenced by genetic drift and gene flow within groups. A smaller proportion of variation among populations suggests

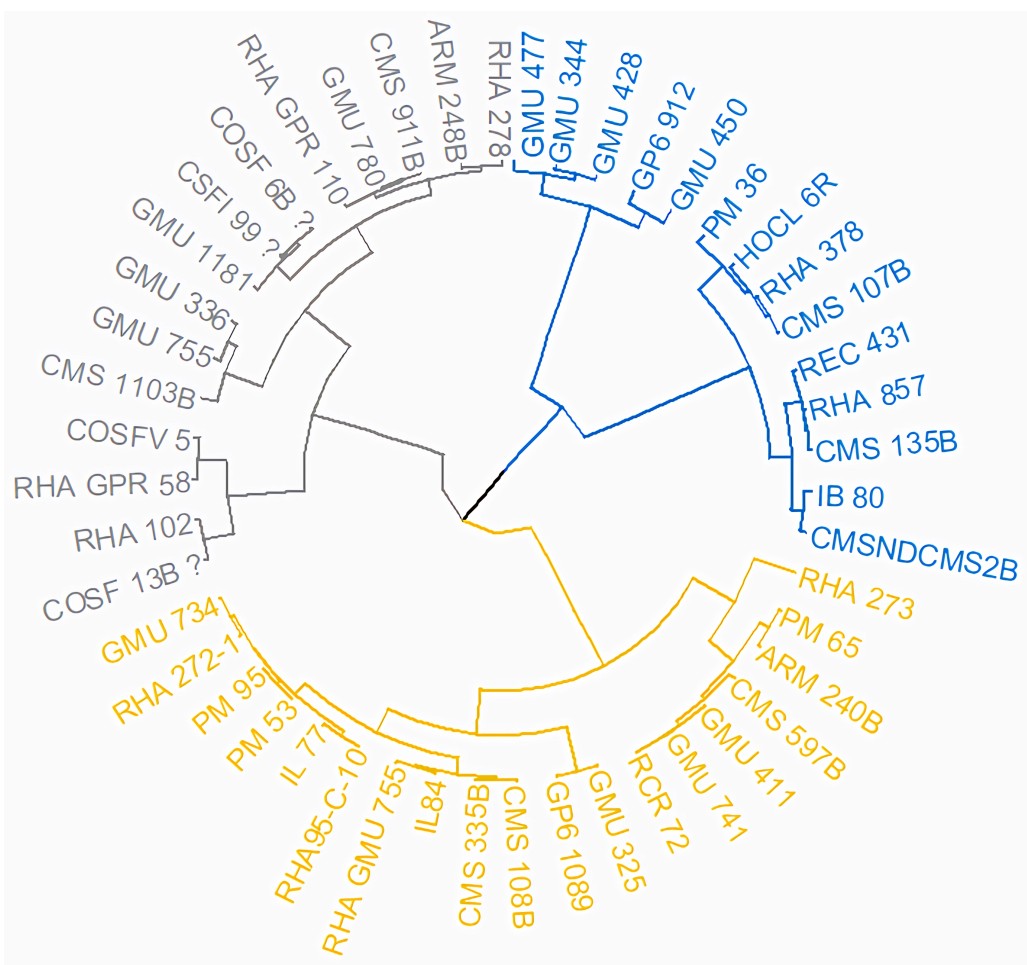

**Figure 14  Grouping of 48 genotypes according to morphological characteristics.** The resulting dendrogram revealed that Cluster I (light yellow) comprised 19 genotypes, Cluster II (light blue) encompassed 15 genotypes, and Cluster III (dark blue) included 14 genotypes.

limited genetic differentiation, possibly due to gene flow, historical migration, or shared ancestry. Similarly, *Gogoi et al. (2023)* found high variation within populations(87%) and comparatively less variation among populations (13%). *Jannatdoust et al. (2016)* observed a higher level of genetic variation within populations (70%) compared to between populations (30%). This higher variation amid specific plants may be attributed to the high allogamous nature of sunflower.

   STRUCTURE uses Bayesian methods to classify individuals into subpopulations and identify consistent subgroups. It reflects genetic interactions, evolutionary history, and gene migration, focusing on population structure, admixture, and genetic clustering. *Mandel et al. (2011)* observed a maximum of $K = 2$ clusters for population structure in 433 sunflower genotypes, while *Ahmed et al. (2022)* found a maximum of $K = 3$, subdividing the population into 3 subpopulations. The genetic structure of plant populations reflects factors such as individual genetics, evolutionary history, and gene migration within and

between species. Nineteen germplasms met the purity threshold (0.80), each subpopulation contained genetically pure germplasms, identified through membership probabilities. Net nucleotide distances among subpopulations varied, with subpopulations II and V having the highest distance, indicating significant genetic differentiation possibly due to factors like geographic isolation or selection pressure. Conversely, subpopulations I and IV showed the lowest distance, suggesting minimal genetic divergence likely facilitated by gene flow or shared ancestry. The fixation index (Fst) measures population differentiation on a scale from 0 to 1. Expected heterozygosity varied: subpopulation IV showed highest diversity, subpopulation III exhibited the lowest diversity with a low Fst value, indicating low genetic differentiation among populations, indicating high gene flow and similar genetic characteristics. Subpopulation IV displayed the highest Fst value, indicating significant genetic differentiation among populations. This implies limited gene flow between populations, resulting in distinct genetic characteristics. (*Patel et al., 2024*) studied subpopulation III shown extreme Fst value (0.3591), indicating significant diversity among subpopulations. *Mandel et al. (2011)* noted pairwise Fst values spanning from 0.016 to 0.183 across 12 cultivar classes. The most significant values were observed between INRA-RHA lines and the remaining gene pool, as well as between RHA-oil lines and other cultivated classifications, suggesting genetic substructure within the germplasm. Genetic distance between populations is crucial for germplasm conservation, with hybrids from genetically distant parents often exhibiting superior traits. Consequently, genetic analysis of population structure delineates breeding subpopulations (*Rauf et al., 2020*). A triangular plot shows individuals' genetic composition across subpopulations, with each corner representing a subpopulation and each point indicating an individual's genetic makeup, visualizing admixture and genetic similarity/dissimilarity. A histogram of alpha values illustrates the distribution of admixture coefficients, showing genome proportions from each subpopulation. A histogram of Fst values displays genetic differentiation between subpopulations, assessing genetic divergence or similarity and providing insights into population differentiation and genetic exchange.

Germplasms were categorized into five populations, with genetic variability parameters. Subpopulation IV displayed the highest values, indicate increased genetic diversity within populations and higher gene flow between populations. Sub population II exhibiting the lowest values, indicate reduced genetic diversity within populations and lower gene flow between populations. Gene flow (Nm), ranging from <1 to >4, shapes genetic diversity. Low flow drives divergence, while high flow fosters uniformity and diversity. A value of 4.01 indicates high flow, promoting genetic homogenization while maintaining diversity. *Gogoi et al. (2023)* observed that Population 1 in Assam showed the highest values for parameters (na, ne, h, I), confirming significant genetic diversity within the population. Moreover, significant genetic diversity was confirmed through total species diversity within the population (Ht) and gene flow (Nm), which were notably higher. *Ibrar et al. (2022)* reported that pairwise genetic distance estimates based on Nei's gene distance index showed the greatest variation between CMS lines and SFP (self-pollinated lines), whereas the lowest diversity was observed between restorer lines and SFP. The high allelic distribution in subpopulation III indicates greater genetic diversity, enhancing adaptability

and resilience. Conversely, the lower allelic distribution in subpopulation IV suggests reduced genetic diversity.

PCA and cluster analysis are essential multivariate tools for genetic diversity exploration, crop evolution tracing, parental line selection, diversity assessment, and environmental interaction. This highlights the substantial explanatory power of the initial four principal components, making them pivotal for understanding parameter relationships. Although the remaining components still contributed to overall variation, their impact on data variability was relatively minor. Genotypes exhibit high diversity and have been utilized across various breeding techniques, positioning them away from the centroid. Conversely, the other genotypes are clustered closer to the centroid, indicating less diversity. The principal components derived from this analysis provide valuable insights for genotype characterization and inform future breeding strategies. *Dudhe et al. (2020)* found that the first four PCA components explained 86.72% of variation, with subsequent components contributing 13.28% or less. *Saux et al. (2020)* observed high PCA percentages in seven clusters, indicating distinct resistance or sensitivity traits. Hierarchical clustering provides clearer genotype grouping, aiding in parent selection for sunflower hybrids (*Ibrar et al., 2024*). Cluster means for seven traits across 2149 entries were recorded (*Dudhe et al., 2020*). In a PCA biplot, the angle between attribute vectors' sines provides an approximate correlation representation. An acute angle (<90°) suggests positive correlation, while a right angle (90°) implies trait independence. However, it's important to note that biplot angles don't directly correspond to correlation coefficients; separate calculation is necessary for accuracy (*Abdi & Williams, 2010*; *Mohi-Ud-Din et al., 2021*; *Antony et al., 2021*). By integrating morphological and molecular marker data with genetic distance analysis, substantial diversity was observed, with genotypes RHA 273 and GMU 325 consistently demonstrating high oil yield per plant. Genotypes GMU 477, GMU 450, COSF 13B, RHA 102, CMS 1103B, and RHA GPR 58 were identified as suitable parents for enhancing oil content in sunflower breeding programs.

## CONCLUSION

Sunflower (*Helianthus annuus* L.) is a globally significant oilseed crop known for its adaptability and high yield potential. This research enhances our understanding of sunflower through insights into genetic divergence, molecular diversity, and selection cues, aiding future breeding efforts to enhance oil production and desired traits. By integrating molecular and morphological data, the study offers crucial insights into the genetic diversity and population structure of sunflower germplasm. SSR markers are effective tools for assessing genetic variability, supporting breeding programs aimed at refining sunflower cultivars for superior traits and productivity. This study utilized 267 loci to characterize genetic diversity across sunflower germplasms, setting the stage for future evolutionary and functional genomic research. Broadening germplasm representation in future studies will further enrich our knowledge of sunflower genetic diversity.

### Funding
This work was supported by the Department of Oilseed, CPBG, TNAU, Coimbatore, Tamil Nadu, India. The funders had no role in study design, data collection and analysis, decision to publish, or preparation of the manuscript.

### Grant Disclosures
The following grant information was disclosed by the authors:
Department of Oilseed, CPBG, TNAU, Coimbatore, Tamil Nadu, India.

### Competing Interests
Sushil Kumar is an Academic Editor for PeerJ.

### Author Contributions
- Sampath Lavudya conceived and designed the experiments, performed the experiments, analyzed the data, prepared figures and/or tables, authored or reviewed drafts of the article, and approved the final draft.
- Kalaimagal Thiyagarajan conceived and designed the experiments, performed the experiments, analyzed the data, prepared figures and/or tables, authored or reviewed drafts of the article, and approved the final draft.
- Sasikala Ramasamy conceived and designed the experiments, performed the experiments, prepared figures and/or tables, authored or reviewed drafts of the article, and approved the final draft.
- Harish Sankarasubramanian conceived and designed the experiments, prepared figures and/or tables, authored or reviewed drafts of the article, and approved the final draft.
- Senthivelu Muniyandi conceived and designed the experiments, prepared figures and/or tables, authored or reviewed drafts of the article, and approved the final draft.
- Anita Bellie conceived and designed the experiments, prepared figures and/or tables, authored or reviewed drafts of the article, and approved the final draft.
- Sushil Kumar analyzed the data, prepared figures and/or tables, authored or reviewed drafts of the article, and approved the final draft.
- Susmitha Dhanapal analyzed the data, prepared figures and/or tables, authored or reviewed drafts of the article, and approved the final draft.

### Data Availability
The raw data is available in the Supplemental Files.

### Supplemental Information
Supplemental information for this article can be found online at http://dx.doi.org/10.7717/peerj.18205#supplemental-information.

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
