# Peer review of "Assessing population structure and morpho-molecular characterization of sunflower (Helianthus annuus L.) for elite germplasm identification"

_PeerJ, doi:10.7717/peerj.18205_

## Round 0.1 · original submission · Major Revisions

· Academic Editor

Major Revisions

Dear Authors

The manuscript cannot be accepted for publication in its current form. It needs substantial revision to meet the journal's standards. The authors are invited to revise the paper, considering all the suggestions made by the reviewers, including the reviewer who rejected it. Please note that requested changes are required for publication.

With Thanks

Reviewer 1 ·

Basic reporting

Here, the authors attempt to use SSR markers to describe the population genetic and morphological diversity of a collection of sunflower germplasm. However, the manuscript discusses little, if any, of the rich literature on comparative genomics and genome-wide association of morphological variation in this group. At the core, please place this work in the context literature to discuss what contributions this work has provided to the greater sunflower or population genetic literature more broadly.

Experimental design

The basic experimental design is fine. However, there is not a sufficient rationale for the author's choices to use approaches that have largely been abandoned in sunflower breeding and population genetics in the early 2010s.

Validity of the findings

While data are provided, the authors do not discuss how this contributes to the literature of sunflower genomics more broadly, and as a reviewer, I fail to see the broader contributions.

Additional comments

41-42: The genus originates from southwestern North America, and the cultivated sunflower was domesticated in central North America.

42-43: what is remarkable about cross-pollination and in relation to the chromosome count?

48: Glucosinolates are lineage-specific metabolites, so they would not be in any Asteraceae, to begin with. This could be improved by comparing other oilseed crops in the context of their evolutionary relationships.

63-66: Consumer sunflowers are planted in homogenous stands, so cross-pollination outside of the US would not be an issue. In most places, the consumer material is an F1 cross, so breeding material is planted in ~2:1 ratios of CMS and fertility-restoring lines. Please clarify what you mean by the statement as written.

67-69: DNA markers have been used in sunflower breeding for over 20 years; please cite any relevant research. This section, along with others, lacks a rationale for using SSR markers over more common approaches. It’s more common to do GBS.

77-79: This needs to be strongly supported. Most of the genome is composed of repetitive elements; however, most traits have oligemic architectures supported by functional biallelic variation. In addition, more extensive population genetic surveys have found better delineation of subpopulations in cultivated sunflowers using SNP data over SSR markers.

93: What is a check variety? Please clarify. Also, where was the germplasm secured from? The discussion suggests INRA and USDA repositories. So, I am curious if these lines have been previously genotyped by either the Mandel, Kane, or Reiseberg groups, which have previously sequenced much of those repositories.

96-98: Please elaborate on cultivation practices, which vary based on location and use type.

127-129: Why was Tang et al. used compared to Mandel et al. 2013, whose primers find ~5000 SSR markers?

Reviewer 2 ·

Basic reporting

no comment

Experimental design

no comment

Validity of the findings

no comment

Additional comments

The manuscript evaluated genetic diversity of forty-eight sunflower genotypes based on morphological traits (9 traits) and molecular markers (267 loci by 103 SSR primers). The experimental design of this study is reasonable and scientific. Many analysis methods for assessing genetic diversity and population structure were adopted and rich research results were obtained. The findings of this study helps to identify potential genetic variations and excellent genes in sunflower, providing a foundation for future sunflower breeding efforts and thus unlocking greater production potential in sunflower. However, there are still many problems in the manuscript that require careful revision. Some of the flaws and corresponding modification suggestions are as follows:
(1) In this manuscript, the subheadings do not meet the standards. For example, the subheading ——“Results”—— is missing from the manuscript; Some subheadings have serial numbers, while others do not. In Line 111, the subheading "Estimation of oil content (%)" appears between "2.3. Morphological characterization" and "2.4 DNA extraction and marker amplification". This is very confusing. In addition, the serial number before the “Discussion” subheading is 4.0. Please carefully correct these errors.
(2) In the results section, there are only some simple descriptive sentences, lacking further analysis. In addition, the first paragraph in 3.1 section (Line167-173) should be deleted because it does not belong to the results of this study. Of course, the authors can also consider placing it in the introduction or discussion section.
(3) The discussion seems to be lengthy and tedious. There are too many overlaps with the results section.
In the discussion section, the authors seem to often adopt a fixed approach: First, introduce the applications and/or advantages of the analysis methods used in the previous results section; then repeat the previous result data of this study, with one or several sentences following each result data, explaining what the result dada indicates; finally, briefly list several research findings using the same analysis method from previous researchers.
Almost every analysis conducted in the previous results section is discussed separately in turn.
Please select and focus on a few meaningful and important issues for in-depth discussion, rather than discussing all research findings. It is better to integrate the results provided by various analyses for comprehensive discussion.
(4) The conclusion is too abstract and vague, lacking specificity. Please rewrite the conclusion again.
(5) It is better to provide the information on the sources of the sunflower used in this study in the supplementary files.
(6) Many of the figures in this manuscript are not yet standardized enough. For example, some sunflower genotype names in Figure 1 are obscured and cannot be seen clearly. The caption of Figure 7 is incomplete. In addition, some figures lack detailed annotation, making it difficult to understand, especially the figures in the supplementary file.
(7) Some tables in the supplementary file are not yet standardized enough. For example, supplementary Table 2 does not provide a sunflower genotype names, only a serial number. Supplementary Table 4 and Supplementary Table 5 should use I-V represent subgroups or subpopulation, instead of Arabic numerals 1-5.

Of course, there are still many other minor flaws (eg.misspelling) in this manuscript that have not been pointed out here. Please revise this manuscript carefully.

·

Basic reporting

There is a lack of clarity in the study and research completed for this article

Experimental design

Data from a single season and one location with unknown 48 sunflower accessions may not be useful.

Validity of the findings

Requires extensive research further confirmation of the identifed material.

Additional comments

Enclosed

---

## Round 0.2 · Major Revisions

· Academic Editor

Major Revisions

Dear Authors

The reviewers have recommended major revisions to your manuscript. Therefore, I invite you to respond to the reviewers' comments and revise your manuscript.

In addition, there are significant concerns about the manuscript's grammar, usage, and overall readability. We, therefore, request that you revise the text to fix the grammatical errors and improve the overall readability of the text.

With Thanks

Reviewer 4 ·

Basic reporting

Abstract:
• The significance of your results needs to be highlighted rather than merely presenting results. What are the implications of your results to the further sunflower breeding needs to be emphasized
Introduction
• This section must highlight why characterisation is needed. It also needs to consider the utility of diversity studies and population structure identification in sunflower breeding.
• Previous studies relevant to the use of SSR markers for studying population structure and diversity need to be highlighted
• What is the research gap in this study
• What is ORS primers???

Materials and methods
• What was the basis for choosing these 48 sunflower germplasms?
• Based on one-season data, is it appropriate to identify high-yielding genotypes for yield and oil, as they are quantitative in nature?
• What was the row length in this study, and what is the number of rows used for experimentation???
Results & Discussion
• What is the difference between population structure and genetic diversity???
• What is the relationship between morphological and molecular diversity in this study??
• No description of morphological data is recorded in the manuscript. Include all the morphological data (per se performance) in the results for better understanding, and it will enhance the quality of the manuscript
• The discussion is currently overly detailed and repetitive, with significant overlap between the content in the results section

Experimental design

Abstract:
• The significance of your results needs to be highlighted rather than merely presenting results. What are the implications of your results to the further sunflower breeding needs to be emphasized
Introduction
• This section must highlight why characterisation is needed. It also needs to consider the utility of diversity studies and population structure identification in sunflower breeding.
• Previous studies relevant to the use of SSR markers for studying population structure and diversity need to be highlighted
• What is the research gap in this study
• What is ORS primers???

Materials and methods
• What was the basis for choosing these 48 sunflower germplasms?
• Based on one-season data, is it appropriate to identify high-yielding genotypes for yield and oil, as they are quantitative in nature?
• What was the row length in this study, and what is the number of rows used for experimentation???
Results & Discussion
• What is the difference between population structure and genetic diversity???
• What is the relationship between morphological and molecular diversity in this study??
• No description of morphological data is recorded in the manuscript. Include all the morphological data (per se performance) in the results for better understanding, and it will enhance the quality of the manuscript
• The discussion is currently overly detailed and repetitive, with significant overlap between the content in the results section

Validity of the findings

Abstract:
• The significance of your results needs to be highlighted rather than merely presenting results. What are the implications of your results to the further sunflower breeding needs to be emphasized
Introduction
• This section must highlight why characterisation is needed. The utility of diversity studies and population structure identification in sunflower breeding also needs to be considered.
• Previous studies relevant to the use of SSR markers for studying population structure and diversity need to be highlighted
• What is the research gap in this study
• What is ORS primers???

Materials and methods
• What was the basis for choosing these 48 sunflower germplasms?
• Based on one-season data, is it appropriate to identify high-yielding genotypes for yield and oil, as they are quantitative in nature?
• What was the row length in this study, and what is the number of rows used for experimentation???
Results & Discussion
• What is the difference between population structure and genetic diversity???
• What is the relationship between morphological and molecular diversity in this study??
• No description of morphological data is recorded in the manuscript. Include all the morphological data (per se performance) in the results for better understanding, and it will enhance the quality of the manuscript
• The discussion is currently overly detailed and repetitive, with significant overlap between the content in the results section

Additional comments

Abstract:
• The significance of your results needs to be highlighted rather than merely presenting results. What are the implications of your results to the further sunflower breeding needs to be emphasized
Introduction
• This section must highlight why characterisation is needed. It also needs to consider the utility of diversity studies and population structure identification in sunflower breeding.
• Previous studies relevant to the use of SSR markers for studying population structure and diversity need to be highlighted
• What is the research gap in this study
• What is ORS primers???

Materials and methods
• What was the basis for choosing these 48 sunflower germplasms?
• Based on one-season data, is it appropriate to identify high-yielding genotypes for yield and oil, as they are quantitative in nature?
• What was the row length in this study, and what is the number of rows used for experimentation???
Results & Discussion
• What is the difference between population structure and genetic diversity???
• What is the relationship between morphological and molecular diversity in this study??
• No description of morphological data is recorded in the manuscript. Include all the morphological data (per se performance) in the results for better understanding, and it will enhance the quality of the manuscript
• The discussion is currently overly detailed and repetitive, with significant overlap between the content in the results section

---

## Round 0.3 · accepted · Accept

· Academic Editor

Accept

Dear Authors,
I am pleased to inform you that the manuscript has improved greatly after the last round of revision and can be accepted for publication.
Congratulations on accepting your manuscript, and thank you for your interest in submitting your work to PeerJ.
With Thanks

Reviewer 4 ·

Basic reporting

Clear and Unambiguos

Experimental design

Appropriate

Validity of the findings

Preliminary results

Additional comments

Improved the manuscript significantly